# Experiential Learning Program to Strengthen Self-Reflection and Critical Thinking in Freshmen Nursing Students during COVID-19: A Quasi-Experimental Study

**DOI:** 10.3390/ijerph17155442

**Published:** 2020-07-28

**Authors:** Yi-Chuan Cheng, Li-Chi Huang, Chi-Hsuan Yang, Hsing-Chi Chang

**Affiliations:** 1Department of Nursing, Asia University, Taichung 41354, Taiwan; chuan70@asia.edu.tw; 2Department of Public Health, China Medical University, Taichung 40402, Taiwan; 3School of Nursing, China Medical University, Taichung 40402, Taiwan; lichi@mail.cmu.edu.tw; 4Department of Nursing, National Taichung University of Science and Technology, Taichung 404, Taiwan; chyang@nutc.edu.tw

**Keywords:** experiential learning program, self-reflection, critical thinking, nurse education

## Abstract

This article focuses on the unique needs and concerns of nursing educators and nursing students in the face of the COVID-19 pandemic. During social distancing, interacting with other human beings has been restricted. This would undermine the experiential learning of nursing students. Hence, it is important to develop and evaluate an experiential learning program (ELP) for nursing education. A pre-test and post-test design were used. The study was conducted in a university in Central Taiwan. A total of 103 nursing students participated in the study from February to June 2019. The study intervention was the experiential learning program (ELP), including bodily experiences and nursing activities with babies, pregnant women, and the elderly. After the intervention, the students completed the self-reflection and insight scale (SRIS) and Taiwan Critical Thinking Disposition Inventory (TCTDI) as outcome measures. An independent *t*-test showed that there was a significant difference between pre-test and post-test in both SRIS and TCTDI (*p* < 0.01). The Pearson product–moment correlation analysis showed that SRIS and TCTDI were significantly positively correlated (*p* < 0.01). ELP has a significant impact on the self-reflection and critical thinking of first-year nursing students, which can be used as a reference for the education of nursing students. During these turbulent times, it is especially vital for faculties to provide experiential learning instead of the traditional teaching concept.

## 1. Introduction

The world is facing unprecedented challenges in the face of a global pandemic. Coronavirus disease 2019 (COVID-19) has, to date, killed thousands worldwide [1]. COVID-19 has also already disrupted universities and academic institutions. Within the health field, schools of nursing are bracing for unique challenges related to our role in helping to develop the next generation of care providers. Social interest in patient safety is growing, and consumers of healthcare services are more likely to obtain healthcare information, which increases the demand for high-quality healthcare [1,2]. Academic nurses have also been quick to adapt in the light of the crisis caused by COVID-19 and many have very quickly moved to online course delivery. Nurse educators and administrators are tasked with ensuring that students meet academic requirements. The most junior nursing students have had their clinical placements postponed due to an imminent shortage of supervisory staff and rapid changes within the clinical environment [3,4,5]. Nursing education is the process of cultivating nursing students to become nursing professionals, and the ability of self-reflection and critical thinking needs to be developed at the stage of freshmen nursing students [6,7,8]. Freshmen nursing students are introduced to clinical assignments to help them begin to develop their nursing clinical imagination, formation of habits of thought, and skilled know-how, and in order to better understand how the prerequisites in the sciences and humanities are relevant to nursing practice, they need to learn simpler aspects of situations before moving on to understanding the whole complex, unfolding clinical situation. This is why situated coaching is essential for the novice, because the novice simply does not have the deep background experiential knowledge yet to recognize the whole clinical situation or make qualitative distinctions within a clinical situation [6,7,8]. Experiential learning is a process of directly recognizing, readily accepting, respecting, and applying the knowledge and abilities currently taught [8,9]. Its operational definition is as follows: experiential learning courses are experiential education. Participants reflect on life and the environment through the process of personal experience and learn the educational method of knowledge accumulation [9].

This article focuses on the unique needs and concerns of nursing educators and nursing students in the face of the COVID-19 pandemic. In this study, the curriculum of ELP activities is designed to guide students to discuss topics and promote their interest in learning through multiple teaching methods and change the classroom learning atmosphere so that students can strengthen their attitudes, abilities, and skills and enhance the core nursing practice while discovering facts and problems, helping them become competent caregivers trusted by their patients and families. 

## 2. Literature Review

### 2.1. Definition and Connotation of Self-Reflection 

The rising tension in nursing is palpable, and, for many of us, this is unprecedented. Reflection is viewed as a learning strategy and a method to improve professional nursing practice [10,11]. Dewey (1993) believed that reflection is a mental activity that has experiential significance [12,13,14,15]. Reflection possesses the following characteristics: inspection, introspection, proactiveness, and thoughtful construction and transformation [16,17]. Transformative learning requires individuals to facilitate their own understanding and experience to restructure or revise their experiential significance, which is used as a direction for future action. In summary, reflection can occur during or after an activity, and it enables the accumulation of personal learning experiences, which facilitate problem-solving [18,19,20]. Self-reflection needs to guide and use structural problems to guide repeated practice. Relevant research reports point out that the biggest advantage of reflective learning is that it can integrate practice and knowledge learning, increase professional responsibilities, increase learners’ knowledge and confidence, increase critical thinking skills, and increase clinical disputes [19,20]. Self-reflection ability is regarded as a learning strategy and a means to improve nursing practice. Self-reflection writing (journal writing) is a common type of reflection writing which is usually carried out after the end of the learning course activities to promote students’ self-understanding of the learning experience, beliefs, and values, thereby transforming learners’ concepts and enhancing self-awareness. Self-reflection writing not only helps students to develop critical thinking skills and learn professional experience but also enhances self-understanding and fosters the ability of nursing staff to respond to emergencies. Reflective diary is the most commonly used in clinical nursing teaching, which can enhance the reflection of students in action and post-action reflection. In addition, diary writing provides a dialogue window between instructor and learner. Therefore, reflective writing is respected by nursing education experts [17,18,19,20].

### 2.2. Definition and Connotation of Critical Thinking

Critical thinking is a form of high-level thinking in which individuals must have an objective attitude to judge the value of things, decide whether they are true or false, good or evil, and draw a conclusion through introspection and logical reasoning. In the English term “critical thinking”, “critical” comes from the root “skeri” and the Greek word “kriterion”. The former refers to cutting, separation, or analysis, while the latter refers to the standard of judgment [13,21]. “Critical thinking” is defined as the psychological process of using concepts, applications, analysis, synthesis, and evaluation techniques to obtain answers or conclusions [21,22]. Therefore, it is necessary for the university teaching community to continue researching active and innovative methodologies to improve the motivation and learning of students. Education in nursing aims to integrate theoretical knowledge into practical knowledge in real-life scenarios and to help students develop their critical thinking [13,22]. Thus, teaching and learning models used in nursing education have changed from traditional models that were solely focused on knowledge transfer to active student learning methods [21,22]. Critical thinking is one of the abilities required by nursing professionals [23]. Nursing professionals can continuously reflect on nursing problems and seek solutions during the nursing process, and nursing professionals need to learn problem solving skills in the face of the drastic changes in the medical environment [13,21,22,23,24]. Teachers can use different teaching strategies in the teaching process to enhance students’ ability to think critically, such as group discussions, clinical practice, writing communication notes, patient simulation, case-based clinical discussions, role-playing, video teaching, and objective structured clinical examination (OSCE), etc., to enhance students’ “critical thinking” and “problem solving” abilities in the hope that students will be able to practice self-awareness and reflection all the time [13,21,22,23].

### 2.3. Definition and Connotation of Experiential Learning 

Experience is the process of experiencing something or a feeling [8,24]. Experiential learning begins with experience, followed by reflection, discussion, and analysis, re-experience and evaluation, and ends with the construction of internalized meaning and value [8,25]. Experiential learning stems from Dewey’s philosophy of learning by doing, which emphasizes learning through participation in reflection and sharing in practice [8,25,26]. It begins with experience, followed by reflection, discussion and analysis, re-experience and evaluation, and ends with the construction of internalized meaning and value [8,14,25].

Dewey established the relationship between learning and experience through reflective observations to assist students in learning by doing [8,15,25,26]. From the nature of education or learning, learning is a process of continuous reorganization and change of experience. It is the conversion process of experience into knowledge, skills, and attitudes, which emphasizes reflection and action as well as the interaction between individuals and situations [8,14,25]. The activity course focuses on integrating life experience, breaking the boundaries of disciplines, and expanding from individuals to the relationships between individuals and society or nature through educational activities of the study fields that embody humanization, lifestyle, adaptability, integration, and modernization. The course focuses on the practical viewpoints of “life experience”, “secondary experience of reflection”, and “to experience by going back to your own life experience”, as well as emphasizing the “reflection” in the continuous reorganization and transformation of experience [8,15,25,26,27]. 

## 3. Methods

### 3.1. Design

This study was a one-group, pre-test–post-test design. Participants had to take the ELP, which was carried out during the semester. The ELP included real intelligent baby care, maternity experience, and elderly experience. After the intervention of ELP, we asked the participants to complete the outcome measures. The details of the intervention are as follows.

#### 3.1.1. Real Intelligent Baby Care (2 Days) 

With the real intelligent baby, students needed to judge when to take care. For example, there were chips in the diapers and baby bottles. When the intelligent babies cried and needed care, the students had to first use their hands to touch the chip induction area, and then comfort, breastfeed, and change diapers according to the needs of the intelligent babies. Each participant brought the smart baby home for two days to immerse themselves in the experiential learning.

#### 3.1.2. Maternity Experience (4 h)

Through the personal experience of wearing a pregnancy dress for 2 h, one can experience the mother’s hard work when she is pregnant. Students can experience the physiological changes produced by the mother during pregnancy, fetal limb pressure, weight gain of 25–30 pounds, and changes in body shape, etc. 

#### 3.1.3. Elderly Experience (4 h)

Students wore assistive appliances for the elderly to experience the deterioration of physical function after age, with the help of creating visual impairment, restricted movement, decreased mobility, reduced sensation of kyphosis, hand sensitivity, experience joint stiffness, fatigue, weakness, as well as physical situations such as a change in mind image and a decline in balance.

After ELP, students had to write journals and participated in a group discussion to share their experiences related to the experiential learning with real intelligent baby care, maternity experience, and elderly experience.

### 3.2. Participants

After the institutional review board approved the study protocol, we recruited a convenience sample of 103 first-year nursing students from a university in Central Taiwan from February 2019 to June 2019.

### 3.3. Measures

The demographic data form was used to collect participants’ characteristics. Outcome measures included the self-reflection and insight scale (SRIS) and the Taiwan Critical Thinking Disposition Inventory (TCTDI).

The self-reflection and insight scale (SRIS) is a Likert scale that assessed the nursing students’ self-reflection with a total of 20 questions and assigned 6 points for each question, ranging from 1 (least matched) to 6 (highly matched). The scale score in total ranges from 20 to 120 points, with higher scores indicating stronger self-reflection ability. Cronbach’s α coefficient for the total scale was 0.79 [19]. The SRIS internal consistency reliability (Cronbach’s α) in the present study was 0.57.

The Taiwan Critical Thinking Disposition Inventory (TCTDI) developed by Yen was used to measure the nursing students’ perception of critical thinking. This scale was composed of 20 questions and had four dimensions: systematicity/analyticity (9 items), open-mindedness (4 items), inquisitiveness (3 items), and reflective thinking (4 items). Each question was answered using a 6-point Likert-type scale ranging from 1 (least matched) to 6 (highly matched), and higher scores indicated higher critical-thinking intention and skill. A previous study noted that the Cronbach’s α of the dimensions ranged from 0.83 to 0.92 [28]. Recent research indicated that Cronbach’s α of the scale was 0.97 [25]. 

### 3.4. Ethical Considerations

The study was approved by the Human Experiment and Committee of Asia University Medical Research Committee. Instructions and a written consent form were provided with the questionnaire and the questionnaire was completed on a voluntary basis. Personal information of each participant was kept confidential by using coded IDs instead of student names. The author’s permission was obtained for the use of SRIS and TCTDI. 

### 3.5. Data Collection

After receiving a verbal briefing on the research and completing informed consent, all participants were asked to complete the demographic data form, SRIS, and TCTDI. One research assistant in each class was invited to assist in participant recruitment and questionnaire distribution. The pre-test data were collected at the semester start date, while the post-test data were collected at the end of the semester (eighteen weeks later). 

### 3.6. Statistical Analysis

SPSS statistical software (Version 19, SPSS Inc., Chicago, IL, USA) was adopted for the analysis. Descriptive statistics included percentage, mean, and standard deviation. Independent *T*-test and the Pearson product–moment correlation analysis were performed. 

## 4. Results

Overall, a total of 103 students participated in this study. Of these, 17 were male (16.50%) and 86 were female (83.50%); most students were at the age of 18 (63.10%). Self-reflection averaged 70.33 points (SD = 7.13) in the pre-test and 72.94 points (SD = 6.80) in the post-test. There was a significant difference between the pre-test and post-test scores in total (t = −2.69, *p* = 0.01) (Table 1). Additionally, there were significant differences between the pre-test and post-test scores in the three domains: behavior, thoughts, and feelings. Critical thinking averaged 86.25 points (SD = 14.65) in the pre-test and 95.41 points (SD = 12.08) in the post-test. There was a significant difference between the pre-test and post-test scores both in total and in each item (Table 2). In addition, there were significant differences between the pre-test and post-test scores in the four domains: systematicity/analyticity, open-mindedness, inquisitiveness, and reflective thinking. The Pearson product–moment correlation analysis showed that the scores of self-reflection and critical thinking were significantly and positively correlated *(p* < 0.001) (Table 3).

## 5. Discussion

This study’s results demonstrated that the total scores of self-reflection and critical thinking had a significantly positive correlation (*p* < 0.001). In the study by Chen and Pai (2018), using the same measures, i.e., TCTDI and SRIS, the authors pointed out that self-reflection and critical thinking would affect each other, although their study sample was a group of nursing students carrying out nursing practicum just before their graduation. However, our findings showed that the ELP had similar effects. In other words, results from both freshmen and students about to graduate demonstrated that self-reflection and critical thinking had positively correlated. 

In our study, the participants’ self-reflection and critical thinking improved after taking the ELP. As the articles appealed, with the actual participation of learners, they can experience the essence of their own learning [29,30]. Additionally, if nursing professionals could perform any nursing practice work in a critical thinking manner, it will solve the needs and nursing problems of patients and ensure the quality of nursing care for patients [23]. 

The theory of experiential learning argues that, during the learning process, if students have personal experiences to internalize their knowledge through cooperation, reflection, questioning, and action, they can apply what they have learned and strengthen their abilities of reflection and critical thinking [30]. In our study, the students shared their personal experiences with classmates after going through the ELP. For example, after caring for a real intelligent baby, students said, “I understood the responsibility of being a mother”; after wearing a pregnancy dress, students shared that “I knew pregnant women’s feeling inconvenience and difficulty”; after wearing assistive appliances for the elderly, students attested that “I felt the feelings of getting losses of eyesight, hearing, teeth health, as well as physical difficulty”. 

The articles showed that ELP creates a real and interactive learning environment. Students can provide feedback and reflection under the guidance of teachers during care activities and can repeatedly learn about knowledge, communication, and critical thinking skills related to nursing, which can enhance students’ experiential learning, boost their self-confidence, and promote their communication and teamwork skills [13,28,29,30].

## 6. Limitations

This study has several limitations. Students’ self-assessment of self-reflection and critical thinking using the questionnaire poses limitations such as subjectivity, although this is the most common form used. In addition, it is uncertain whether the difference between pre-test and post-test comes entirely from the course since a control group is absent in this study.

## 7. Impact Statement

This research suggested that schools at all levels incorporate ELP into the learning process so that students can strengthen their attitudes, abilities, and skills in their learning activities.

## 8. Conclusions

This study manifested that self-reflection and critical thinking might share a core, which resulted in the significantly positive correlation between SRIS and TCIDI. The nature of experiential learning is that students learned through active participation, gaining knowledge and insights. Our intervention, the curriculum of ELP activities, helped the students to increase their abilities of self-reflection and critical thinking. As the ELP program can be performed without facing people, it is suitable for use during a pandemic.

## Figures and Tables

**Table 1 ijerph-17-05442-t001:** Comparison of scores of SRIS before and after ELP (*n* = 103).

Domains of SRIS	Mean		t	*p*
	Pre-Test	Post-Test		
I. Behavior	20.59	21.25	−1.98	0.04 *
II. Thoughts	21.39	22.04	−1.60	0.11
III. Feelings	28.35	29.65	−2.53	0.00 **
Total	70.33	72.94	−2.69	0.01 **

* *p* < 0.05, ** *p* < 0.01, *** *p* < 0.001.

**Table 2 ijerph-17-05442-t002:** Comparison of scores of TCTDI before and after ELP (*n* = 103).

Domains of TCTDI	Mean		t	*p*
	Pre-Test	Post-Test		
I. **Systematicity/analyticity**	38.62	42.74	4.42	0.00 ***
II. Open-mindedness	17.01	18.87	4.43	0.00 ***
III. Inquisitiveness	13.20	14.50	−3.95	0.00 ***
IV. Reflective thinking	17.42	19.29	−4.30	0.00 ***
Total	86.25	95.41	−4.62	0.00 ***

* *p* < 0.05, ** *p* < 0.01, *** *p* < 0.001.

**Table 3 ijerph-17-05442-t003:** Correlation between reflection and critical thinking (*n* = 103).

Variables	SRIS	TCTDI
SRIS	1.00	0.30 *
TCTDI	0.30 *	1.00

* *p* < 0.05, ** *p* < 0.01, *** *p* < 0.001.

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
