# Peer review of "Experiential Learning Program to Strengthen Self-Reflection and Critical Thinking in Freshmen Nursing Students during COVID-19: A Quasi-Experimental Study"

_ijerph, 2020, doi:10.3390/ijerph17155442_

Round 1

Reviewer 1 Report

Line

Comment/Recommendation

56

I don’t know if it is cultural, but I prefer nursing professional to nursing staff. Nursing staff for me includes not just RNs but LPNs, nurse aids and technicians. Being an RN is more than just technical skills and I believe it is the professional RN student that this article is about.

56-57

Please explain further why there is a need to develop self-reflection and critical thinking skills as a freshman. Why not as juniors or some other level for example.

81

The subheading for this paragraph is on the previous page line 80 and is easily missed and confusing.

118

In the design, could you further explain what was the actual ELP and reflective interventions? Types of nursing experiences are mentioned (babies, OB, geriatrics) but what did the students actually do for reflection? Did they do reflection in practice while providing patient care, with questions provided by faculty for example? This usually addresses cognitive areas of learning and would make sense for the critical thinking part of this research.

Or reflection on practice such as journaling after completing patient care and when students are alone and have time? This is often geared more towards affective changes and works toward finding meaning in actions and experiences to be used later. I’m really curious as to what was the process of reflection used.  In practice and on practice are 2 different ways of reflecting and often have different outcomes but are related, both based on experiential learning or life experiences.

155

The scale is confusing, is 1 actually matched? Or unmatched?

213

Summarize the first 2 sentences as this is a repeat of lines 171…

242 and 254

Repetitive sentences. I found others throughout the manuscript. Things can be summarized better, reducing your word count and making your presentation more concise and clear for the reader.

283 and Conclusion

Conclusion. This is the first time you have mentioned COVID and health crisis. Need to present this earlier in the manuscript and relate it to EPL and reflection.

The first paragraph does not flow logically, referring to the experiences of faculty, the needs of patients, and less on students. This section needs more work.

323

References: Only 10/53 (18.9%) of references are 5 years or less (from 2015 -2020). The majority seem to be 10 years old or older. Need to update references and lit review.

This study has distinct potential with some reworking.

Statistics and results were sound.

I found the literature review needing updating.

The manuscript in general needs more editing for several reasons, one being sentence structure and wording, the other being redundant information (looks like it was literally cut and pasted into other parts of the manuscript in some cases)

I would like to see something added about the reflective process.

Thank you.

Author Response

Reviewer’s Comments (Reviewer #1)

Authors’ Response

Thank you for the excellent comments.

Comments and Suggestions for Authors:

Line

Comment/Recommendation

56

I don’t know if it is cultural, but I prefer nursing professional to nursing staff. Nursing staff for me includes not just RNs but LPNs, nurse aids and technicians. Being an RN is more than just technical skills and I believe it is the professional RN student that this article is about.

Thank you for the valuable comment. We have changed “nursing staff” to “nursing professional”.

56-57

Please explain further why there is a need to develop self-reflection and critical thinking skills as a freshman. Why not as juniors or some other level for example.

Thank you for the valuable comment. In order to evaluate the improvement of self-reflection and critical thinking after the four years nursing education, we have to keep the baseline of self-reflection and critical thinking of the students.

81

The subheading for this paragraph is on the previous page line 80 and is easily missed and confusing.

Thank you for your suggestion. We have been amended on line line88. (page3)

118

In the design, could you further explain what was the actual ELP and reflective interventions? Types of nursing experiences are mentioned (babies, OB, geriatrics) but what did the students actually do for reflection? Did they do reflection in practice while providing patient care, with questions provided by faculty for example? This usually addresses cognitive areas of learning and would make sense for the critical thinking part of this research.

Thank you for your suggestion. More details have been added to clarify on line line129-152. (page3-4)

Or reflection on practice such as journaling after completing patient care and when students are alone and have time? This is often geared more towards affective changes and works toward finding meaning in actions and experiences to be used later. I’m really curious as to what was the process of reflection used.   In practice and on practice are 2 different ways of reflecting and often have different outcomes but are related, both based on experiential learning or life experiences.

Thank you for your suggestion. More details have been added to clarify on line line129-152. (page3-4)

155

The scale is confusing, is 1 actually matched? Or unmatched?

Thank you for the valuable comment. It has been revised on line line157-171. (page4)

213

Summarize the first 2 sentences as this is a repeat of lines 171…

Thank you for your suggestion. It has been revised on line207. (page 4)

242 and 254

Repetitive sentences. I found others throughout the manuscript. Things can be summarized better, reducing your word count and making your presentation more concise and clear for the reader.

Thank you for your suggestion.It has been revised.

283 and Conclusion

Conclusion. This is the first time you have mentioned COVID and health crisis. Need to present this earlier in the manuscript and relate it to EPL and reflection.

Thank you for your suggestion.It has been added on line line33-62. (page1-2)

323

References: Only 10/53 (18.9%) of references are 5 years or less (from 2015 -2020). The majority seem to be 10 years old or older. Need to update references and lit review.

Thank you for your suggestion.Articles are added on page 7-8.

1.     Covid, C. D. C., & Team, R. (2020). Severe outcomes among patients with coronavirus disease 2019 (COVID-19)—United States, February 12–March 16, 2020. MMWR Morb Mortal Wkly Rep, 69(12), 343-346.

2.     Park, M., Jeong, M., Lee, M., & Cullen, L. (2020). Web-based experiential learning strategies to enhance the evidence-based-practice competence of undergraduate nursing students. Nurse Education Today, 104466. https://doi.org/10.1016/j.nedt.2020.104466

3.     Jackson, D., Bradbury‐Jones, C., Baptiste, D., Gelling, L., Morin, K., Neville, S., & Smith, G. D. (2020). Life in the pandemic: Some reflections on nursing in the context of COVID‐19. Journal of clinical nursing.

4.     Swift, A., Banks, L., Baleswaran, A., Cooke, N., Little, C., McGrath, L., ... & Williams, G. (2020). COVID‐19 and student nurses: A view from England. Journal of Clinical Nursing.

5.     Hayter, M., & Jackson, D. (2020). Pre‐registration undergraduate nurses and the COVID‐19 pandemic: students or workers?. Journal of Clinical Nursing.

6.     Powers, K., Herron, E. K., & Pagel, J. (2019). Nurse preceptor role in new graduate nurses' transition to practice. Dimensions of Critical Care Nursing, 38(3), 131-136.

7.     Fagan, J. M., & Coffey, J. S. (2019). Despite challenges: Nursing student persistence. Journal of Nursing Education, 58(7), 427-430.

8.     Grace, S., Stockhausen, L., Patton, N., & Innes, E. (2019). Experiential learning in nursing and allied health education: Do we need a national framework to guide ethical practice?. Nurse education in practice, 34, 56-62.

9.     Akella, Devi. "Learning together: Kolb's experiential theory and its application." Journal of Management & Organization 16.1 (2010): 100-112. https://doi.org/10.1017/S1833367200002297

Reviewer 2 Report

ABSTRACT:

clarify in the background the articulation between a program that apparently already exists and the covid 19 pandemic phase. Explain the program in the bacground. 
- It is necessary to better articulate the contents in the objective / design / method subsections.

- the first phrase of results, should be in methods. 

INTRODUCTION: 

- Clarify firts time use of "ELP".

- It is not clear the changes and challenges to education, caused by the covid 19 and, if this program already existed, what adaptations were necessary to face the difficulties caused by the pandemic.

METHODS:

  • the topics and expeciences of the Program, should be explained in the introduction, not in method.
  • No data colection is presented. All the authors have exposed is the theoretical context of the program. How did you collect data?
  • On 3.6 validity and reliability, authors also are talkink about "materials". Values about reliability shoul be presented refering that the alpha values are from previous studies.

RESULTS: 

  • t has to be explained to which comparison the value of t student in table 1 refers.
  • review the articulation with the tables. The sentence starting on line 188 and the one starting on line 195, do not seem to refer to table 2. on line 175 authors present the reference table and join 1,3 - stricter articulation with the text is suggested
  • when exposing the differences item by item in table 2, they should indicate whether the differences are statistically significant and better describe these results in the text.

DISCUSSION

  • On line 222 are presented and discussed data that ar not in results.
  • data between line 223 to 229 are not discution but backgound.
  • data between line 230 to 246 mus be linked to what this means to nursing students and nursing education. 
  • Between 258 to 274 authors are nort discussion their on article.
  • In general readers want to Know what change with the ptoblem and why. Authors must look their on data and discuss.

CONCLUSIONS

- little is concluded from your findings. A lot of what you have is background and you even put things that help them better understand the program

# References throughout the text and on all reference list must be revised to link with journal guidelines (ex: on introduction: line 87, 250, )

Author Response

Reviewer’s Comments (Reviewer # 2)

Authors’ Response

Thank you for the excellent comments.

Comments and Suggestions for Authors:

ABSTRACT

clarify in the background the articulation between a program that apparently already exists and the covid 19 pandemic phase. Explain the program in the bacground.

- It is necessary to better articulate the contents in the objective / design / method subsections.

- the first phrase of results, should be in methods.

Thank you for your suggestion. More details have been added to clarify on line line15-29. (page1)

INTRODUCTION

- Clarify firts time use of "ELP".

- It is not clear the changes and challenges to education, caused by the covid 19 and, if this program already existed, what adaptations were necessary to face the difficulties caused by the pandemic.

Thank you for your suggestion. It has been added to clarify on line line33-62. (page1-2)

METHODS

the topics and expeciences of the Program, should be explained in the introduction, not in method.

No data colection is presented. All the authors have exposed is the theoretical context of the program. How did you collect data?

On 3.6 validity and reliability, authors also are talkink about "materials". Values about reliability shoul be presented refering that the alpha values are from previous studies.

Thank you for your suggestion. It has been added to clarify on line line129-187. (page3-5)

RESULTS

t has to be explained to which comparison the value of t student in table 1 refers.

review the articulation with the tables. The sentence starting on line 188 and the one starting on line 195, do not seem to refer to table 2. on line 175 authors present the reference table and join 1,3 - stricter articulation with the text is suggested

when exposing the differences item by item in table 2, they should indicate whether the differences are statistically significant and better describe these results in the text.

Thank you for your suggestion. It has been added to clarify on line line189-199. (page5)

DISCUSSION

On line 222 are presented and discussed data that ar not in results.

data between line 223 to 229 are not discution but backgound.

data between line 230 to 246 mus be linked to what this means to nursing students and nursing education.

Between 258 to 274 authors are nort discussion their on article.

In general readers want to Know what change with the ptoblem and why. Authors must look their on data and discuss.

Thank you for your suggestion.It has been revised on line207-225. (page6)

CONCLUSIONS

- little is concluded from your findings. A lot of what you have is background and you even put things that help them better understand the program

# References throughout the text and on all reference list must be revised to link with journal guidelines (ex: on introduction: line 87, 250, )

Thank you for your suggestion.It has been revised on line235-238. (page6)

Round 2

Reviewer 2 Report

Results: format the tables

Discussion: Must be improved

-The data in the first paragraph (line 509) need the bibliographic reference cited and can be reinforced with comparison with other studies.
- In the second paragraph (line 212-213), the findings need to be discussed. If the instruments have different dimensions, it will be relevant to discuss them and compare them with other studies.

- The findings from line 217-221 can frame the entire discussion. How do the dimensions of what was measured show the growth of this experience?

- The statement on line 222-226, which is a recent study, can be improved (the study is from 2013).

Conclusion:

- It can be reinforced, It can be extended to the nature of learning through experience, from the data

References:

- throughout the text must be revised to link with journal guidelines in some parts

Author Response

Reviewer’s Comments (Reviewer # 2)

Authors’ Response

Thank you for the excellent comments.

Comments and Suggestions for Authors:

Results

format the tables

Thank you for your suggestion.It has been revised on line200-205. (page5-6).

Discussion: Must be improved

The data in the first paragraph (line 209) need the bibliographic reference cited and can be reinforced with comparison with other studies.

Thank you for your suggestion. It has been revised on line 208-213 (page 6).

In the second paragraph (line 212-213), the findings need to be discussed. If the instruments have different dimensions, it will be relevant to discuss them and compare them with other studies.

Thank you for your suggestion. It has been revised on line 214-218(page 6)

- The findings from line 217-221 can frame the entire discussion. How do the dimensions of what was measured show the growth of this experience?

Thank you for your suggestion. More details have been added on line line223-227 (page6).

The statement on line 222-226, which is a recent study, can be improved (the study is from 2013).

Thank you for your suggestion.It has been revised on line228-232. (page6)

Conclusion

- It can be reinforced, It can be extended to the nature of learning through experience, from the data

It has been added on line line243-244. (page7)

References

- throughout the text must be revised to link with journal guidelines in some parts

Thank you for your suggestion.It has been revised on line253-314. (page7-8)
